# Comparison of Peritoneal Carcinomatosis Scoring Methods in Predicting Resectability and Prognosis in Gynecologic Malignancies

**DOI:** 10.3390/jcm10122553

**Published:** 2021-06-09

**Authors:** María Teresa Climent, Anna Serra, Juan Gilabert-Estellés, Juan Gilabert-Aguilar, Antoni Llueca

**Affiliations:** 1Department of Obstetrics and Gynecology, University General Hospital of Castellón, 12004 Castellón, Spain; serraa@uji.es (A.S.); llueca@uji.es (A.L.); 2Department of Medicine, University of Jaume I, 12071 Castellón, Spain; juangilaeste@yahoo.es (J.G.-E.); juangilabertaguilar@gmail.com (J.G.-A.); 3Department of Obstetrics and Gynecology, University General Hospital of Valencia, 46014 Valencia, Spain

**Keywords:** laparoscopy, peritoneal carcinomatosis, PCI, Fagotti index, resecability

## Abstract

Objective: Peritoneal carcinomatosis is a disease’s presentation in the advanced stages of many gynecologic tumours. The distribution and volume of the disease are the main factors in achieving complete debulking. Diagnostic laparoscopy is a technique to allow evaluation of the disease. This study’s objective is to compare two laparoscopic scores (Fagotti’s index and Sugarbaker’s peritoneal cancer index (PCI)) and assess the diagnostic accuracy to select patients for neoadjuvant treatment and reduce unnecessary laparotomies. Methods: A non-randomised retrospective cohort study was conducted in patients with peritoneal carcinomatosis (ovarian and endometrial origin) who underwent laparoscopy and subsequent laparotomy. We evaluated the scores’ ability to predict incomplete surgery and whether they were related to the patients’ prognosis. Results: We included 34 patients, of which 23.5% received neoadjuvant chemotherapy. The rate of complete cytoreductive surgery was 79.4% (*n* = 27 patients). The highest sensitivity was obtained with a PCI value greater than 20. It was the best parameter to determine incomplete debulking. Survival curves were analysed according to the “cut off” established for each score, and statically significant differences were found using PCI with respect to Fagotti’s Index. However, these differences were not found with Fagotti’s score. Conclusion: The best diagnostic method to classify patients with peritoneal cancer is the PCI. It could be adapted to each surgical team because it allows identifying the “cut off point”, which depends on incomplete surgery rate.

## 1. Introduction

Ovarian cancer is a significant cause of death from gynaecologic cancer, with 70–80% of cases diagnosed in advanced stages [1,2]. Complete primary debulking surgery is an independent survival factor [1,2] because the tumour residue’s size is inversely related to overall survival [3,4].

The distribution and volume of the disease are the main factors in achieving complete debulking. The FIGO classification is not a useful tool to determine the disease’s extent due to patients’ disparity with the same stage.

Among the diagnostic methods, laparotomy continues to be the most accurate one to determine the burden of disease in patients. It is an invasive method of diagnosis with associated complications [1].

Diagnostic laparoscopy is a less-invasive evaluation alternative that reduces suboptimal surgery from 39% to 10% [2,5]. This technique can improve optimal surgery rates, limiting unnecessary morbidity and reducing costs derived from unnecessary laparotomy [6,7].

The application of an evaluation method that allows quantifying the disease in patients with peritoneal carcinomatosis would improve the test’s accuracy.

The use of scores is a method to compare results and determine in an objective way the treatment indicated for each patient; however, it is currently a challenge.

The use of laparoscopic scores could reduce unnecessary laparotomies for those patients to whom the possibility of complete cytoreduction is scarce or non-existent. Neoadjuvant therapy and interval surgery are presented as valid alternatives for these patients [2].

The most widely used laparoscopic predictive models evaluating ovarian cancer’s extension are the peritoneal carcinomatosis index, described by Sugarbaker, and the Fagotti score [8].

Traditionally, gynecologist oncologists in Europe have used Fagotti’s index to predict cytorreductive surgery in laparotomy for gynecologic malignancies [9,10,11].

This study aims to compare these scores, assessing the diagnostic accuracy to select patients for neoadjuvant treatment and reduce unnecessary laparotomies, morbidity and mortality derived from them, and correlation with the prognosis of these patients.

## 2. Material and Methods

### 2.1. Patients

115 patients with diagnosis of peritoneal carcinomatosis.

Patients were required to meet all the following eligibility criteria:Aged 18 years old or older.Diagnosis of peritoneal carcinomatosis.Written consent of surgery.

We excluded patients who did not have diagnosis of peritoneal carcinomatosis or did not give consent for data collection.

71 patients (61.7%) had not undergone laparoscopy at diagnosis, so they were excluded from the study.

Of 54 patients on whom laparoscopy was performed to evaluate primary debulking, 10 (18.5%) were referred for neoadjuvant treatment without subsequent laparotomy.

A total of 81 patients were referred for neoadjuvant treatment for reasons attributable to the patient (e.g., comorbidity, anaesthetic risk), to the disease (e.g., metastasis, number of intestinal resections) or, criteria of unresectability established by the Multidisciplinary Unit of Abdominal Pelvic Surgery (MUAPOS) (Radiologic criteria; lung metastasis, hepatic metastasis in three or more hepatic segments, severe hepatic pedicle involvement and progression after neoadjuvant chemotherapy) [1,3,6].The comparison was made among patients where the laparoscopy score suggested the possibility of performing a successful debulking surgery, and subsequent laparotomy was performed within a maximum period of 10 days (34 patients) (Figure 1).

The absence of tumour disease was defined as complete surgery, not distinguishing between optimal (tumour residue less than 1 cm) and suboptimal surgery (tumour residue greater than 1 cm). We included these two types of surgery under the term ‘incomplete cytoreductive surgery’ [4].

The patients’ preoperative evaluation consisted of a complete gynaecological examination, transvaginal ultrasound, tumour markers (Ca 125, CEA, Ca 19.9 and HE4), complete analysis, and thoracic-abdominopelvic computed tomography. Informed consent was obtained from all patients, and the ethics committee approved the study.

### 2.2. Surgical Technique

A 15 mm supraumbilical longitudinal incision was made, and a 12 mm Hasson’s trocar was introduced in the abdomen. A suprapubic 5 mm accessory trocar was placed, adding another trocar 5 mm in the iliac fossa if required, both under direct vision.

The entire abdominal cavity was examined, and a biopsy of the ovaries, metastatic nodules, or peritoneal surface was performed to confirm the diagnosis.

In our unit, it is considered that PCIs greater than 20 are associated with high morbidity with a decrease in the possibility of complete debulking. The value considered as unresectable in the case of the Fagotti index is higher than 8 [12,13]. However, the subjective evaluation of the oncologist surgeon, the histology of the disease, and the performance status of the patient are evaluated prior to the decision to perform surgery for posterior debulking.

If the patient is considered to be subject to debulking surgery, it was performed in the second stage, after histological diagnosis, with a xipho-pubic midline incision.

### 2.3. Scores

#### 2.3.1. Fagotti Score

The Fagotti Score is based on the evaluation of seven parameters: omental cake, peritoneal carcinomatosis, diaphragmatic carcinomatosis, mesenteric retraction, stomach infiltration, bowel infiltration, and liver metastases. Each parameter is valued with a 0 if absent or 2 if present. The total value is between 0 and 14. A value above or equal to 8 is related to suboptimal surgery [7,12].

#### 2.3.2. Sugarbaker Score (PCI)

The abdomen is divided into nine regions: central (0), right hypochondrium (1), left hypochondrium (3), epigastrium (2), left flank (4), left iliac fossa (5), pelvis (6), right iliac fossa (7), and right flank (8). Four regions corresponding to the digestive tract are added: upper jejunum (9), lower jejunum (10), upper ileum (11), and lower ileum (12). Each area scores 0 if there is no evidence of a tumour; 1 with a tumour smaller than 0.5 cm, 2 with a tumour up to 5 cm, and 3 with a tumour larger than 5 cm or confluent.

The value obtained is between 0 and 39.

The “cut off” established in the bibliography is between 10 and 20 [5,6].

The description of the scores is described in Appendix A.

### 2.4. Statical Analysis

The objective of this retrospective descriptive non-randomised study is to compare the two laparoscopic scores to determine sensitivity, specificity, positive predictive value (PPV), negative predictive value (NPV), and diagnostic accuracy to predict suboptimal cytoreduction in patients with gynaecological peritoneal carcinomatosis.

The demographic characteristics of each patient, the tumour, and the surgical results obtained were analysed.

Parameters such as FIGO stage, PCI, and Fagotti score were determined for each patient who underwent laparoscopy and laparotomy.

The main objective was to determine if the models correlated with the incomplete debulking rate (optimal or suboptimal) and overall survival, using ROC (receiver operating characteristic) curves and the Kaplan–Meier method for survival.

Significance was assumed with a *p*-value less than 0.05.

The statistical program used was IBM SPSS Statistics version 19 (IBM Corp, released in 2010. IBM SPSS Statistics for Windows, Version 19.0. Armonk, NY, USA).

## 3. Results

Between 2013 and 2017, 115 patients were diagnosed with peritoneal carcinomatosis at the University General Hospital of Castellón. The data were extracted from the database of the Multidisciplinary Unit of Abdominal Pelvic Surgery (MUAPOS). During the study period, 34 patients who underwent laparoscopy surgery and posterior cytoreductive surgery were included in the data analysis.

The mean age of the patients was 62 years.

According to the FIGO classification [1,2], 79.4% (*n* = 27 patients) were classified as FIGO III, 20.6% (*n* = 7 patients) presented stage IV.

58.8% were high-grade serous tumours.

The origin of the peritoneal carcinomatosis was ovarian in 85.3%; in 14.7% of the patients, it was endometrial.

The mean tumour marker (Ca 125) was 1020.94 IU/mL.

23.5% of the patients received neoadjuvant chemotherapy.

Maneuvers were performed in the upper abdomen in 61.8% of the patients.

The complete cytoreduction rate was 79.4% (*n* = 27 patients) with 52.9% postoperative complications, of which 5.8% were severe.

The suboptimal cytoreduction rate was 20.6%, leaving a macroscopic tumour residue equal or more than 1 cm.

The clinicopathological features shown in the table (Table 1).

Overall survival at 5 years was 46.31 months (95% CI 33.06–69.57) (Figure 2).

The mean of Fagotti’s score with laparoscopic approach was 4.79 (se 0.406), and Sugarbaker’s score was 13.03 (se 1.06).

A comparison of means between the two scores and their capacity to predict complete debulking applied by laparoscopy or laparotomy was conducted.

Statistically significant differences were found in the means of the Sugarbaker index in complete cytoreductive surgery (R0) about incomplete cytoreductive surgery (No R0) and not at all in the Fagotti’s score (Table 2 and Table 3).

The predictive capacity of the different indices applied laparoscopically to predict incomplete debulking surgery was evaluated.

The highest sensitivity was obtained with PCI value greater than 20, resulting in 43%, with a specificity of 88%, a PPV of 50% and an NPV of 86%.

Based on the data provided, the Sugarbaker score was the best parameter to determine the possibility of incomplete debulking (Table 4).

The most significant discriminative capacity test was the peritoneal cancer index described by Sugarbaker, regardless of whether it was applied via laparoscopy or laparotomy (Figure 3).

This data was evaluated using the ROC curves.

Survival curves were analysed according to the “cut off” established for each score evaluated and determined if they had a prognostic impact when this score was applied via laparoscopy.

These tables show the overall survival, depending on whether the value of the index applied via laparoscopy is greater than the established values 10 and 20 for the Sugarbaker PCI, and 8 for the Fagotti index.

These differences in overall survival were found with statistically significant differences when using the PCI (Figure 4 and Figure 5 and Table 5 and Table 6); however, these differences were not found with the Fagotti’s score (Figure 6 and Table 7).

To determine the risk of incomplete debulking, the relative risk was calculated for the scores described by the “cut off.”

A laparoscopic-PCI >20 was a risk factor for incomplete debulking and unnecessary laparotomy RR 3.5 95% CI (10.44–11.734).

This association was not evidenced with PCI cut-off of 10, nor with the Fagotti score.

## 4. Discussion

The present study shows that the peritoneal carcinomatosis index described by Sugarbaker for colorectal cancer is a valid diagnostic test that provides prognostic value in evaluating peritoneal carcinomatosis of gynaecological origin.

Patients with a PCI greater than 20 showed a decrease of 37.61 months in overall survival [6,14].

The results obtained in our sample for Fagotti’s index in specificity and positive predictive value present values below those described in their study, showing an AUC of 0.66, also lower than that obtained in the external validation of Brun et al. and Chéreau, which was 0.74 and 0.69, respectively [6,8,10,11,12] These findings are probably a consequence of the suboptimal cytoreduction rate since in Fagotti’s validation study, it was 32.8%, while in this sample, the rate is 20.6% [10]

The best diagnostic test to determine the eligible patients for neoadjuvant treatment is the Sugarbaker’s peritoneal cancer index, increasing diagnostic accuracy with the increase in the cut-off point.

This cut-off point should be established according to the complete debulking rate obtained in each working group [13].

One of the factors that influence overall survival is the volume of the disease.

The article by Llueca et al. [5,6,13] shows that the subgroup of patients with PCI less than or equal to 10 presents better survival than a value above 20. These findings are confirmed in our study with a decrease in survival as the peritoneal cancer index value increases. This prognostic impact is not observed in Fagotti’s model because it is not a model that quantifies disease.

The gold standard for evaluating the burden of peritoneal carcinomatosis is laparotomy. In an article by Llueca et al. [5], two predictive models were compared; the accuracy of this evaluation increases with the laparotomic exploration [9,15,16]. However, due to the morbidity associated with this technique, it is not considered a diagnostic technique. [17]. However, the potential discrepancy between the PCI assessment by laparoscopy and laparotomy could interfere with the main objective of the study [5,15,18].

Nevertheless, the objective of the study is to establish a better index to apply the laparoscopic method to identify with the least error the patients who would benefit from neoadjuvant therapy because complete debulking surgery is unlikely [19,20].

One of the strengths of this study is the homogeneity of the sample. According to the same criteria for unresectability, all patients were evaluated and treated by the same surgical team with an established suboptimal cytoreduction rate (tumour residue greater than 1 cm) of approximately 16%.

Another noteworthy point of this study is to include patients with peritoneal carcinomatosis of endometrial origin. This subgroup has a worse prognosis as it is a tumour with less chemosensitivity than the ovary, and neoadjuvant treatment in this group is not standard clinical practice [9,21].

The study’s main limitation is a retrospective evaluation of the data. A selection bias was achieved when choosing the patients who underwent laparoscopy and later laparotomy. This fact implies that the surgical team has already selected the patients as probably resectable. This is the reason why there are few patients with PCI values greater than 20 and few patients with suboptimal surgery. Another limitation is the inclusion of patients undergoing neoadjuvant therapy (23.5%). This patient usually has a lower burden of disease; however, when subjected to laparoscopy before surgery, they probably have a greater volume of disease greater than expected in the pre-surgical study [3,22].

One of the weakness of the study is the small sample size, but there is a lack of information in this field, and we could contribute to closing the gap.

It is necessary to carry out prospective studies with a greater number of patients to verify the results obtained in our evaluation.

## 5. Conclusions

Based on the data obtained in our study, it is possible to suggest that the best diagnostic method to classify patients with peritoneal cancer to indicate primary treatment is the peritoneal carcinomatosis index. It presents a better correlation of the extension of the disease and provides prognostic information.

We consider this score could be adapted to the suboptimal cytoreduction rate of each surgical team, allowing the cut-off point to be established at a particular moment and evolving according to the results obtained by the surgical team.

## Figures and Tables

**Figure 1 jcm-10-02553-f001:**
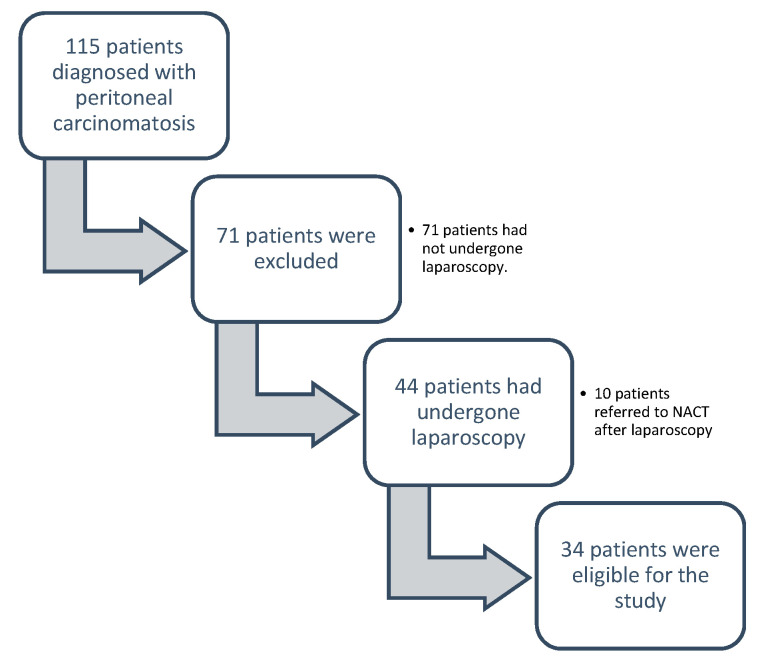
Patient selection.

**Figure 2 jcm-10-02553-f002:**
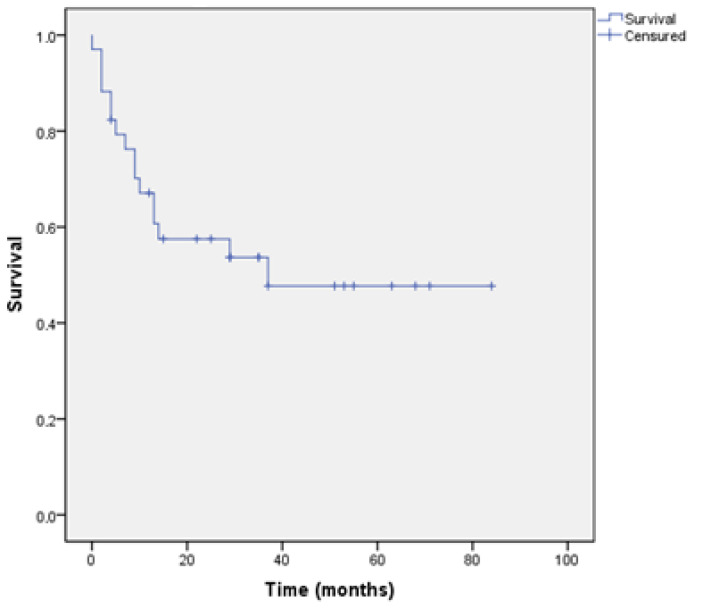
Overall Survival.

**Figure 3 jcm-10-02553-f003:**
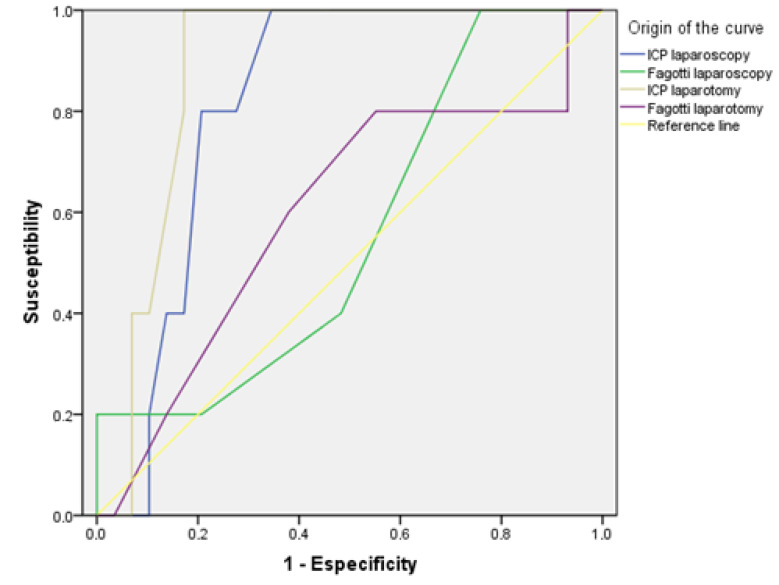
Roc curves shows the accuracy of these tests applied for laparotomy or laparoscopy.

**Figure 4 jcm-10-02553-f004:**
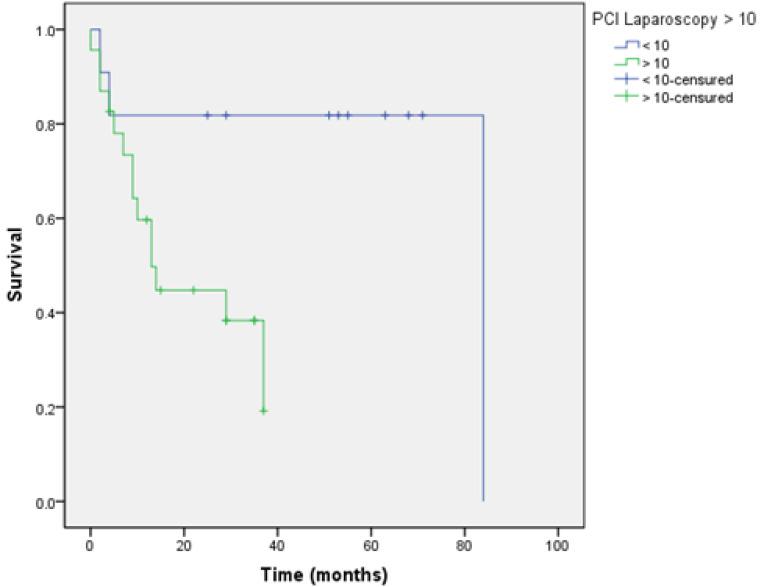
Survival PCI > 10.

**Figure 5 jcm-10-02553-f005:**
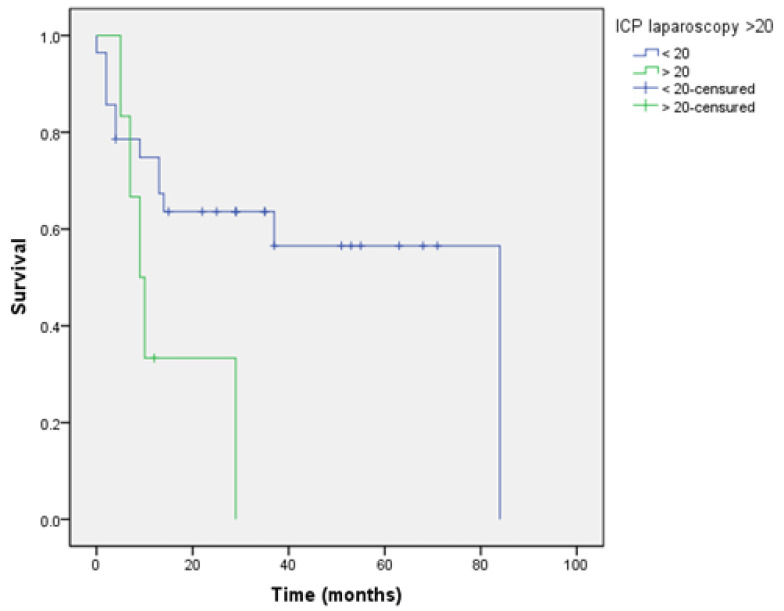
Survival PCI > 20.

**Figure 6 jcm-10-02553-f006:**
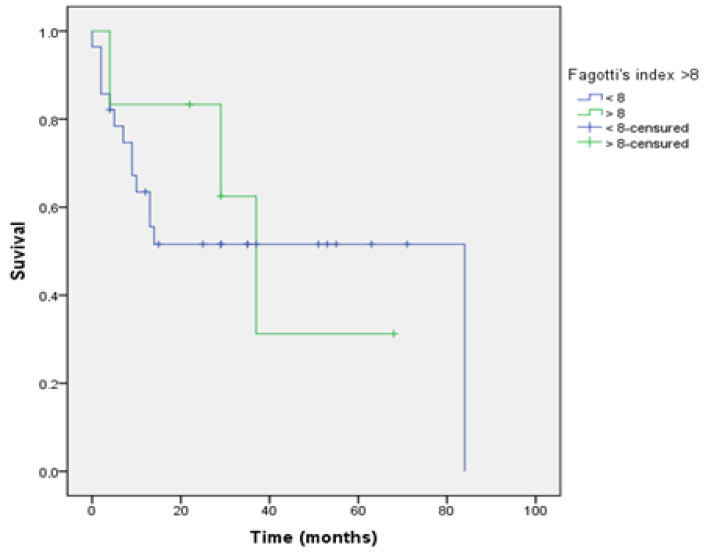
Survival Fagotti’s Index.

**Table 1 jcm-10-02553-t001:** Clinicopathological features.

	*n* = 34	%
Median age at surgery (years)	61.97	
Neoadjuvant chemotherapy	8	23.5
Surgical procedures upper abdomen	20	58.8
Exitus	17	50
Mean follow-up time (months)	25.79	
Mean pre-surgical Ca 125	1020.94	
Complete cytoreductive surgery	27	79.4
Complete/optimal cytoreductive surgery	29	85.3
Suboptimal cytoreductive surgery	7	20.6
FIGO stage		
III	27	79.4
IV	7	20.6
Histology		
Serous	20	58.8
Endometroid	7	20.6
Others	7	20.6
Tumour origin		
Ovarian	29	85.3
Endometrial	5	14.7
Tumour		
Primary	31	91.2
Relapse	3	8.8

**Table 2 jcm-10-02553-t002:** Laparoscopic index and complete cytoreductive surgery.

Laparoscopy	R0	No R0	*p*
PCI	11.44 (se 1.13)	19.14 (se 1.16)	0.002
Fagotti	4.78 (0.46)	5.71 (se 0.808)	0.338

R0: complete cytoreductive surgery. No R0: incomplete cytoreductive surgery. SE: standard error.

**Table 3 jcm-10-02553-t003:** Laparotomic index and complete cytoreductive surgery.

Laparotomy	R0	No R0	*p*
PCI	10.30 (se 1.16)	23.86 (se 1.43)	0.000
Fagotti	5.56 (se 0.59)	7 (se 1.09)	0.273

R0: complete cytoreductive surgery. No R0: incomplete cytoreductive surgery.

**Table 4 jcm-10-02553-t004:** Diagnostic Parameters.

Laparoscopic Score	Sensibility	Specificity	PPV	NPV	Accuracy
PCI ≥ 10	30%	88%	30%	100%	53%
PCI ≥ 20	43%	88%	50%	86%	79%
Fagotti ≥ 8	14%	81%	16%	78%	68%

PPV: positive predictive value. NPV: negative predictive value.

**Table 5 jcm-10-02553-t005:** Survival PCI “cut off” 10.

	Mean Overall Survival (Months)	95% CI
PCI LPS < 10	69.27	46.66–91.89
PCI LPS ≥ 10	20.22	13.82–26.61

PCI LPS: Laparoscopic Peritoneal Carcinomatosis Index.

**Table 6 jcm-10-02553-t006:** Survival PCI “cut off” 20.

	Mean Overall Survival (Months)	95% CI
PCI LPS < 20	52.44	37.35–67.54
PCI LPS ≥ 20	14.38	5.76–23.90

PCI LPS: Laparoscopic Peritoneal Carcinomatosis Index.

**Table 7 jcm-10-02553-t007:** Survival Fagotti “cut off” 8.

	Mean Overall Survival (Months)	95% CI
Fagotti < 8	47.76	31.54–61.97
Fagotti ≥ 8	39.52	19.25–59.71

## Data Availability

The data presented in this study are available on request from the corresponding author. The data are not publicly available due to privacity of the patients included in this article.

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
