# Peer review of "Comparison of Peritoneal Carcinomatosis Scoring Methods in Predicting Resectability and Prognosis in Gynecologic Malignancies"

_jcm, 2021, doi:10.3390/jcm10122553_

Round 1

Reviewer 1 Report

Dear Authors

I read with attention your manuscript.

The sentence "The best diagnostic method to classify patients with peritoneal cancer is the PCI." used to conclude the abstract is wrong. PCI is not a diagnostic method but a severity score.

Please, when a revised version is submitted, highlight the modifications.

Regards

Author Response

REVIEWER 1:

Dear Authors

I read with attention your manuscript.

The sentence "The best diagnostic method to classify patients with peritoneal cancer is the

PCI." used to conclude the abstract is wrong. PCI is not a diagnostic method but a severity

score.

Please, when a revised version is submitted, highlight the modifications.

Regards

Dear Reviewer

Thank you very much for the correction made.

I appreciate your interest.

I agree with you.

We use the Sugarbaker’s peritoneal carcinomatosis index (PCI) like a diagnosis method during

the diagnostic laparocopy perform. PCI help us to decide the primary treatment.

But, as you say the PCI is, also, a severity and prognosis score.

We’ll specify it in introduction section L 53 – 56:

“This study aims to compare these scores, assessing the diagnostic accuracy to select patients

for neoadjuvant treatment and reduce unnecessary laparotomies, morbidity and mortality

derived from them and correlation with the prognosis of these patients”

Our group, Multidisciplinary Unit of Abdomino – Pelvic Oncologic Surgery (MUAPOS) published

an article to stablish the prognostic value of Sugarbaker’s PCI.

  1.  

Llueca A, Escrig J, Serra-Rubert A, Gomez-Quiles L, Rivadulla I, Játiva-Porcar R, et al.

Prognostic value of peritoneal cancer index in primary advanced ovarian cancer. Eur J

Surg Oncol. 2018;44(1):163–9.

In Page 8, in the discusión section L210 – 212 “The present study shows that the peritoneal

carcinomatosis index described by Sugarbaker for colorectal cancer is a valid diagnostic test

that provides prognostic value in evaluating peritoneal carcinomatosis of gynaecological

origin” we try to explain the prognostic and severity value associated to this index.

We’ll specify it better in the manuscript.

I hope I have made the appropriate corrections.

Reviewer 2 Report

I understand the premise of assessing the quantity of carcinoma prior to debulking surgery. This article indicates in a very small population that PCI is superior. 

My reservations are that this work is inherently biased by failure to control for the different treatment algorithms these patients have. The methodology ios adequate, however, the sample size is small. 

Additionally authors do not justify the correct approach to these patients and do not suggest whether these methods at laparoscopy are superior to imaging modalities alone. 

I think the rationale is a reasonable one, and PCI is widely used already, therefore, do not feel the findings are sufficiently novel.

Author Response

REVIEWER 2:

I understand the premise of assessing the quantity of carcinoma prior to debulking surgery. This article indicates in a very small population that PCI is superior.

My reservations are that this work is inherently biased by failure to control for the different treatment algorithms these patients have. The methodology ios adequate, however, the sample size is small.

Additionally authors do not justify the correct approach to these patients and do not suggest whether these methods at laparoscopy are superior to imaging modalities alone.

I think the rationale is a reasonable one, and PCI is widely used already, therefore, do not feel the findings are sufficiently novel.

Dear Reviewer

Thank you very much for the correction made.

I appreciate your interest.

We understand that the sample is small, but we consider there is a lack of information in this field, and we could contribute it.

However, we are working to perform a prospective study to try to evaluate this point, and we hope to publish in this magazine.

Traditionally, gynecologist oncologist in Europe have used Fagotti’s index to predict the cytorreductive surgery in laparotomy.

In Introduction section, L 51 – 52: “Traditionally, gynecologist oncologist in Europe have used Fagotti’s index to predict the cytorreductive surgery in laparotomy in gynecologic malignancies. “

Our group have worked with Sugarbaker’s peritoneal carcinomatosis index (PCI)

We use a algorithm treatment of our working work, Multidisciplinary Unit of Abdomino – Pelvic Oncologic Surgery (MUAPOS).

We evaluate our patients through computed tomography (TC) by intravenous contrast and oral contrast to evaluate the radiologic carcinomatosis peritoneal index. In patients with a correct performance status and doubts about the possibility of resecability we perform a laparoscopy, independently of type of primary tumour or moment of surgery (primary debulking sugery or Interval debulking surgery)

In Methods section, L72 – 75 “criteria of unresectability established by the Multidisciplinary Unit of Abdominal Pelvic Surgery (MUAPOS) (Radiologic criteria; lung metastasis, hepatic metastasis in three or more hepaticsegments, severe hepatic pedicle involvement and progression after neoadjuvant chemotherapy)”

The effort of our group has been to implement PCI, not only as a quantifier of the amount of abdominal disease, but also as a prognostic factor for patients with gynecological cancer, predominantly ovarian cancer.

We facilitate a different publications about this topic, above belongs our group.

  1. Llueca A, Escrig J, Serra-Rubert A, Gomez-Quiles L, Rivadulla I, Játiva-Porcar R, et al. Prognostic value of peritoneal cancer index in primary advanced ovarian cancer. Eur J Surg Oncol. 2018;44(1):163–9.
  1. Llueca A, Serra A, Rivadulla I, Gomez L, Escrig J, Játiva-Porcar R, et al. Prediction of suboptimal cytoreductive surgery in patients with advanced  ovarian cancer based on preoperative and intraoperative determination of  the peritoneal carcinomatosis index.  World J Surg Oncol. 2018;16(1):1–7. 
  1. Llueca A, Serra A, Delgado K, Maiocchi K, Jativa R, Gomez L, et al. A radiologic- laparoscopic model to predict suboptimal (Or complete and optimal) debulking surgery in advanced ovarian cancer: A pilot study. Int J  Womens Health. 2019;11:333–42. 
  1. Llueca A, Serra A, Herraiz JL, Rivadulla I, Gomez-Quiles L, Gilabert-Estelles J, et al. Peritoneal carcinomatosis index as a predictor of diaphragmatic involvement in stage III and IV ovarian cancer. Onco Targets Ther.  2018;11:2771–7.

In gynecological oncological diseases, as previously comented, the main index used to assess resectability is the Fagotti Index.

There is a few bibliography that compares these two index to assess the correlation with the extension of the disease and the prognosis. We consider it is a strong point of our article.

However, we know that the sample is small, we’ll discuss it in the article, as a weakness of it.

In discusión section; L 258 – 261:“One of the weakness of the study is the small sample size, but there is a lack of information in this field, and we could contribute it. It is necessary to carry out prospective studies with a greater number of patients to verify the results obtained in our evaluation.”

We hope that you consider our article approppiate to publish in your magazine.

Round 2

Reviewer 1 Report

Thank you for your reply.

Regards

Reviewer 2 Report

I appreciate the author's reply and appreciate a l;ack of information in this area. With some english corrections and taking into account the views of other reviewers I can agree to accept the paper

This manuscript is a resubmission of an earlier submission. The following is a list of the peer review reports and author responses from that submission.

Round 1

Reviewer 1 Report

This manuscript deal with an interesting topic that is gaining more and more importance in the choice of the best treatment for patients with ovarian cancer.  The article highlights the importance of using an appropriate peritoneal carcinomatosis scoring system in order to avoid useless laparotomy. 

Major comments:

  • bibliography should be implemented
  • In section 2.3.2 about Peritoneal Cancer Index a "cut off" extablished in the bibliography is cited: could the authours explain more in detail the meaning of this sentence??
  • In the Results section the authors described that 7 FIGO IV patients were submitted to explorative laparoscopy; would the authors explain the need of this kind of surgery in patients presenting extra-abdominal or metastatic disease (therefore suitable for neoadjuvant chemotherapy)?
  • In terms of completeness of cytoreductive surgery, it could be useful to distinguish CC-0, CC-1 and CC-2 patients.
  • In table 2  the authors reported 17 patients as exitus? Is it correct? Did really 50% of patients pass away during the intervention or in the following week???   
  • In table 2 the authors reported as complete cytoreductive surgery 27 patients but in the following line the number of complete/optimal cytoreduction (i think it's a synonim or is it a different category?) is reported to be 29. Please clarify this point.
  • It's really hard to understand the meaning of Figure 2 that is probably one of the cornerstone of the results; an explanation of the figure could help reader to catch the meaning. 

Minor comments:

  • table n. 1 is not useful for supporting the study patients' enrollment.
  • In section 2.3 the authors described Fagotti score as composed by 7 variables, but only 6 of them are reported.
  • the authors sometimes referred to PCI as ICP.

Author Response

Response to Reviewer 1 Comments

Thank you very much for the correction made.

I appreciate your interest.

I hope I have made the appropriate corrections.

Point 1: bibliography should be implemented

Response 1: We’ll complete the bibliography.

Point 2: In section 2.3.2 about Peritoneal Cancer Index a "cut off" extablished in the bibliography is cited: could the authours explain more in detail the meaning of this sentence??

Response 2:

In our oncolgical group; Multidisciplinary Unit Abdomino – Pelvic oncological surgery (MUAPOS) the PCI value to avoid the surgery is up to 20.

This value could change depends on the surgical team and the associated morbility.

In the bibliography the limit value to surgery is between 10 and 20. It depends on the surgical group.  

The bibliographic references of our publications are attached in case you wish to consult them:

  1. Llueca A, Escrig J, Serra-Rubert A, Gomez-Quiles L, Rivadulla I, Játiva-Porcar R, et al. Prognostic value of peritoneal cancer index in primary advanced ovarian cancer. Eur J Surg Oncol. 2018;44(1):163–9.
  2. Llueca A, Serra A, Rivadulla I, Gomez L, Escrig J, Játiva-Porcar R, et al. Prediction of suboptimal cytoreductive surgery in patients with advanced ovarian cancer based on preoperative and intraoperative determination of the peritoneal carcinomatosis index. World J Surg Oncol. 2018;16(1):1–7.

Point 3: In the Results section the authors described that 7 FIGO IV patients were submitted to explorative laparoscopy; would the authors explain the need of this kind of surgery in patients presenting extra-abdominal or metastatic disease (therefore suitable for neoadjuvant chemotherapy)?

Response 3:

In table 1 we expose the criteria of unresecability established by our Multidisciplinary Unit Abdomino – Pelvic oncological surgery (MUAPOS).  (Llueca A, Serra A, Rivadulla I, Gomez L, Escrig J, Játiva-Porcar R, et al. Prediction of suboptimal cytoreductive surgery in patients with advanced ovarian cancer based on preoperative and intraoperative determination of the peritoneal carcinomatosis index. World J Surg Oncol. 2018;16(1):1–7.)

We considered that the patients with citological positive pleural effussion or parenchymal liver  metastasis lower tan 3 have a surgical option. Any metastasis that does not compromise the life of the patient in the following 6 months is a candidate for an assessment of a surgical option

In this cases we used the laparoscopy to assess the patients we had doubts about resecability.

Point 4: In terms of completeness of cytoreductive surgery, it could be useful to distinguish CC-0, CC-1 and CC-2 patients.

Response 4:

We use only two options to classify the surgery.

The recent bibliography establish that the objetive of surgery in gynecological tumours is to achieve complete cytorreduction, that means, absence of macroscopic residual tumour.

We talk about only two options;

- RO (complete cytorreductive surgery); absence of macroscopic tumour

- No R0 (no complete cytorreductive surgery): residual macroscopic tumour of any size.

We did this specification in subsection of Material and Methods in L77 - 78. (Llueca A, Serra A, Delgado K, Maiocchi K, Játiva-Porcar R, et al. A radiologic-laparoscopic model to predict suboptimal (or complete and optimal) debulking surgery in advanced ovarian cancer: a pilot study.Int Jour of Women Health. 2019;11 333–342.)

We are agree with you that we must include this specifications in surgical technique and use these nomenclature (complete and no complete cytorreductive surgery) in subsequent subsections.

Point 5: In table 2  the authors reported 17 patients as exitus? Is it correct? Did really 50% of patients pass away during the intervention or in the following week???  

Response 5:

This is a huge mistake. Is the total exitus during follow – up of the patients.

It is referred to the total sample.

In the subsection about the results, we stablished that severe complications is about 5.8%, but we didn’t have any death in the postoperative.

We have made the arrengement for this.

My apologies for this serious mistake.

(P4 Table 1).

Point 6: In table 2 the authors reported as complete cytoreductive surgery 27 patients but in the following line the number of complete/optimal cytoreduction (i think it's a synonim or is it a different category?) is reported to be 29. Please clarify this point.

Response 6:

In gynecological tumours the complete surgery is the term that we use for a cytorreductive surgery without macroscopic tumour residue.

Optimal surgery is used when the tumor residue is less tan 1 cm.

This is the explanation of the difference about the number of patients in this two groups.

Point 7: It's really hard to understand the meaning of Figure 2 that is probably one of the cornerstone of the results; an explanation of the figure could help reader to catch the meaning.

Response 7:

The analysis of ROC curves (receiver operating characteristic curve) is a statistical method to determine the diagnostic accuracy of these tests, being used for three specific purposes:

- to determine the cut-off point of a continuous scale at which the most sensitive and specificity.

- evaluate the discriminatory capacity of the diagnostic test.

- compare the discriminative capacity of two or more diagnostic tests that express their results as continuous scales.

This figure shows the diagnostic accuracy of each of the tests applied via laparoscopic or laparotomic. The most powerful test is PCI applied by laparotomic and laparoscopic.

We could explain this figure in the results. We try to explain in the line 57 to 59, but it was probably confuse. We’ll improve this explanation.

Thank you very much for this contribution, because this figure is one of the strengths of the study

(P6 Figure 2)

Minor comments:

Point 1: Table n. 1 is not useful for supporting the study patients' enrollment.

Response 1:

Good observation.

We included this table to try to explain the criteria of unresecability, and patients who meet these criteria in the imaging test are not subjected to a diagnostic laparoscopy and are referred for neoadjuvant treatment.

If you consider that it is not useful, it will be withdrawn in the next version. (P2)

Point 2: In section 2.3 the authors described Fagotti score as composed by 7 variables, but only 6 of them are reported.

Response 2:

Thank for this appreciation

We miss the bowel infiltration.  (P3 L 106 – 107)

Point 3: The authors sometimes referred to PCI as ICP.

Response 3:

It’s another mistake. We have resolved it in the new edition.

Reviewer 2 Report

Dear Author,

I rewieved the manuscript entitled “Comparison of Peritoneal Carcinomatosis Scoring Methods in Predicting Resectability and Prognosis in Gynecologic Malignancies”.

This is a retrospective analysis of 34 patients treated for ovarian or endometrial peritoneal carcinomatosis, aiming at comparing the prognostic value of two scores (Sugarbaker’s PCI and Fagotti’s score) obtained preoperatively by laparoscopy, to predict a complete resection.

The question is relevant however, the article suffers from several flaws.

The English should be revised.

Abstract : please give precise results rather than “we found differences using PCI”

The presence of endometrial cancer patients in the cohort should be specified in the abstract

Introduction

P1 l39-42: the meaning of these sentences is unclear

Methods

The beginning of the methods part is actually results. the study population inclusion criteria should be described in the methods and the selected population should be described in the results, ideally supported by a flowchart.

Please specify in the selection process which was the histology (ovarian or other) for each mentioned sub-groups.

P2 l69-70 : please define optimal and sub-optimal surgery even if both were included : what was the size of the maximal residue tolerated in your cohort ? Moreover that concept is used also p3l110-111 in the evaluation criteria of the study.

P2 l78-79 : trocar in the right iliac fossa should be avoided. It is linked to a significant risk of parietal tumor seeding, demonstrated in a lot of publications. That kind of parietal attempt has been reported as an independent risk of poor survival in several studies. International guidelines regarding staging laparoscopy in the field of peritoneal carcinomatosis recommend the placement of all the trocars on the abdominal midline.

Results

P3l125 : Means should be given with standard error.

Numbers reported in the text should indicate each time what a number refers to ?

Discussion

The potential discrepancy between the PCI assessment by laparoscopy and laparotomy should be discussed as it could interfere with the main objective of the study (cf Passot G et al Brit J Surg 2016)

Author Response

Response to Reviewer 2 Comments

Dear Author,

I rewieved the manuscript entitled “Comparison of Peritoneal Carcinomatosis Scoring Methods in Predicting Resectability and Prognosis in Gynecologic Malignancies”.

This is a retrospective analysis of 34 patients treated for ovarian or endometrial peritoneal carcinomatosis, aiming at comparing the prognostic value of two scores (Sugarbaker’s PCI and Fagotti’s score) obtained preoperatively by laparoscopy, to predict a complete resection.

The question is relevant however, the article suffers from several flaws.

Dear Reviewer,

Thank you very much for your appreciation.

We agree the study presents various biases, however it is a good approximation to try to determine the best rate of laparoscopic peritoneal carcinomatosis to try to improve the selection of patients for primary treatment in gynecological cancer.

We hope that the corrections made allow us to highlight the strengths of the study.

Point 1: The English should be revised.

Response 1:

As you can see in the acknowledgments, the article has been reviewed by Begoña Bellés Fortuño, PhD, is a senior lecturer in the Department of English Studies at Universitat Jaume I, Spain. She is a senior. She was a Morley Scholar in the ELI (English Language Institute) at the University of Michigan (Ann Arbor, USA) in 2007, where she worked together with the MICASE team. She has been a visiting researcher in Universities such as Midsweden University (Sweden), Katolische Univesität Eichstaett (Germany) or the University of Edinburgh (UK).

She can provide you with a certificate of editing the article if you require it.

Point 2

Abstract : please give precise results rather than “we found differences using PCI”

The presence of endometrial cancer patients in the cohort should be specified in the abstract

Response 2:

If you agree, we can affirm in the abstract that “statistically significant differences were found in favor of the use of the PCI respect to the Fagotti index.”

 We don’t include in the abstract the presence of endometrial cancer patients, because it is the 14.7% of the sample and the effect in the results we think that it’s small. Nevertheless, we agree that. It ‘s a small sample, and it’s probably that 5 cases of endometrial cancer it’s interesting in the study design. We have made a correction about this point.

(P1 L18 – 20)

Introduction

Point 3:

P1 l39-42: the meaning of these sentences is unclear

Response 3:

In this sentence we try to explain the limitation to use the exploratory laparotomy to evaluate the extent of the disease, because it’s an invasive technique with complications that could  delay the begining of the primary treatment of the patients.

 We re-write this lines to achieve a better understanding. (P1 L37 – 42)

Methods

Point 4:

The beginning of the methods part is actually results. the study population inclusion criteria should be described in the methods and the selected population should be described in the results, ideally supported by a flowchart.

Response 4:

You are right.

In P2 L59 – 64) we try to specified the inclusion criteria.

It’s probably the explantion to choose the sample, it’s probably difficult. We agree that the flowchart is useful to simplify this subsection.  (P3)

We recolocate the selected population to results.

Point 5:

Please specify in the selection process which was the histology (ovarian or other) for each mentioned sub-groups.

Response 5:

We didn’t differ between the tumour origin or histology, because we treated the peritoneal carcinomatosis regardless it’s origin

Point 6:

P2 l69-70 : please define optimal and sub-optimal surgery even if both were included : what was the size of the maximal residue tolerated in your cohort ? Moreover that concept is used also p3l110-111 in the evaluation criteria of the study.

Response 6:

In the P2 L77 – 79 we explain the type of surgery.

We evaluated a complete or no complete surgery (optimal or suboptimal).

- Complete (R0): is defined as absence of macroscopic residue tumour.

- No complete; is any residue tumour, regardless of the size. We didn’t differ between optimal and sub-optimal in this paper.

We change the optimal or sub – optimal surgery to no complete cytorreductive surgery.

We re – write this terms to avoid confussion, because it is an important focus of the study.

(P3 L88 – 90)

Point 7

P2 l78-79 : trocar in the right iliac fossa should be avoided. It is linked to a significant risk of parietal tumor seeding, demonstrated in a lot of publications. That kind of parietal attempt has been reported as an independent risk of poor survival in several studies. International guidelines regarding staging laparoscopy in the field of peritoneal carcinomatosis recommend the placement of all the trocars on the abdominal midline.

Response 7:

We have analyzed port - site - metastasis in our population (patients with peritoneal carcinomatosis), and the incidence of port - site metastasis does not exceed 2%.

In any case, in the laparotomic surgery the port-site of the previous laparoscopy is always sectioned.

However, there are some articles that it wasn’t found an increase of port-site-metastasis.

We totally agree.

However, we only use this trocar in cases that we need another laparoscopic forceps to inspect the abdominal cavity properly. We try to avoid the use of this trocar.

  • Ataseven B, Grimm C, Harter P, Heikaus S, et al. Prognostic Impact of Port-Site Metastasis After Diagnostic Laparoscopy for Epithelial Ovarian Cancer. Annals of Surgical Oncology. 2016. 23, 834–840
  • Lago V, Gimenez L, Matute L, Padilla-Iserte P, et al. Port site resection after laparoscopy in advance ovarian cancer surgery: Time to abandon?. Surg Oncol. 2019. 29:1-6)

Results

Point 8:

P3l125 : Means should be given with standard error.

Numbers reported in the text should indicate each time what a number refers to ?

Response 8:

This numbers were referred to the mean of different index; PCI and Fagotti’s index.

In the table 4 and 5, the value of this index in complete cytorreductive surgery (R0) and incomplete cytorreductive surgery (No R0).

The appropriate corrections have been made so the understanding

Thanks for this appreciation.

(P5 L 162 – 167; P6 L174 – 175)

Discussion

Point 9:

The potential discrepancy between the PCI assessment by laparoscopy and laparotomy should be discussed as it could interfere with the main objective of the study (cf Passot G et al Brit J Surg 2016)

Response 9:

The gold standard for evaluating the burden of peritoneal carcinomatosis is laparotomy. As demonstrated in our study, in which two predictive models are compared, the accuracy of the evaluation increases if the laparotomic exploration of the cavity is included. However, due to the morbidity associated with this technique, it is not considered as a possible diagnostic technique.

We associate the bibliographic reference in case it is useful

Moreover, the objetive of the study is to stablish the better index to apply laparoscopic way to identifty the patients who would benefit from neoadjuvant therapy as complete debulking is unlikely.

It isn’t a comparison between laparoscopic o laparotomic approach. This is an interesting additional information.

Thanks for this suggestion, it is very useful and we will use it in the discussion section to emphasize about the diagnostic value of laparoscopy.

(Llueca A, Serra A, Delgado K, Maiocchi K, Játiva-Porcar R, et al. A radiologic-laparoscopic model to predict suboptimal (or complete and optimal) debulking surgery in advanced ovarian cancer: a pilot study.Int Jour of Women Health. 2019;11 333–342.)

 (P9 L243 – 253)

Round 2

Reviewer 1 Report

I truly believe that the topic is very interesting and that the identification of a laparoscopic score for the staging of peritoneal carcinomatosis is a crucial point for subsequent therapeutic evaluations.

However, I believe that the description of the study sample is still not clear. Furthermore, in the patient selection flow chart, the topic is not clarified. It starts from 115 patients, from which 61 are removed: 54 and 81 patients remain on the same level: 135 patients??

Further consideration: it is often stressed along the paper that the size of the tumor residue is one of the most statistically significant factors inversely related to the overall survival. 

To my observation concerning the residual disease (point 4) you reply that your evaluation foresees a dichotomous classification: complete (R0) or not complete cytoreductive surgery (any size).
But within Table 2 there are two different denominations: complete / optimal or suboptimal cytoreductive surgery. So does suboptimal cytoreduction indicate any macroscopic residual disease regardless of size? and the optimal cytoreduction can include a residual disease <1 cm as indicated by you in point 6?
I believe that the question, which is decisive for the paper, needs to be clarified.

Author Response

Point 1: I truly believe that the topic is very interesting and that the identification of a laparoscopic score for the staging of peritoneal carcinomatosis is a crucial point for subsequent therapeutic evaluations.

However, I believe that the description of the study sample is still not clear. Furthermore, in the patient selection flow chart, the topic is not clarified. It starts from 115 patients, from which 61 are removed: 54 and 81 patients remain on the same level: 135 patients??

Response 1: Thanks for your observation.

We try to explain in this flowchart the method to choose the sample.

We only analyze the patients with laparoscopy and posterior laparotomy.

It’s a mistake, the correct number is 71, not 81.

71 patients refer to the total of the sample that send to neadjuvant treatment, 61 don’t perform laparoscopy, and 10, we perform laparoscopy but they not eligible to posterior laparotomy.

We’ll change the flowchart.

(Page 2: L71 – 76,  Page 3: Flowchart 1)

Point 2:

Further consideration: it is often stressed along the paper that the size of the tumor residue is one of the most statistically significant factors inversely related to the overall survival. 

To my observation concerning the residual disease (point 4) you reply that your evaluation foresees a dichotomous classification: complete (R0) or not complete cytoreductive surgery (any size).
But within Table 2 there are two different denominations: complete / optimal or suboptimal cytoreductive surgery. So does suboptimal cytoreduction indicate any macroscopic residual disease regardless of size? and the optimal cytoreduction can include a residual disease <1 cm as indicated by you in point 6?
I believe that the question, which is decisive for the paper, needs to be clarified.

Response 2:

I appreciate your interest.

In ovarian cancer, we use the next terms; ´

- Complete cytoreductive surgery; absence macroscopic residue tumour.
- Optimal cytoreductive surgery; residual disease < 1 cm
- Suboptimal cytoreductive surgery; residual disease >/= 1 cm

“SEGO Oncoguide: Epithelial Cancer of the ovary, tube and peritoneum 2014. Clinical practice guidelines in gynecological and breast cancer. SEGO Publications, October 2014. "

We considered that it was interesting to establish in the table the percentage of complete, complete and optimal surgery with respect to the percentage of suboptimal surgery. Because the discussion raises the possibility of adapting the value of the score to the different suboptimal debulking rates of the surgical teams.

However, we included in the manuscript an aclaration about this point in table 2.

(Page 5, L 157 – 161)
